# Removal of Bisphenol S (BPS) by Adsorption on Activated Carbons Commercialized in Brazil

**DOI:** 10.3390/ijerph21060792

**Published:** 2024-06-18

**Authors:** Nayara dos Santos Oliveira, Yovanka Perez Ginoris, Harsha Ratnaweera

**Affiliations:** 1Department of Civil and Environmental Engineering, Faculty of Technology, University of Brasília, Brasília 70910-900, Brazil; yovanka.perez@gmail.com; 2Department of Building and Environmental Technology, Faculty of Science and Technology, Norwegian University of Life Science, NO-1432 Ås, Norway; harsha.ratnaweera@nmbu.no

**Keywords:** adsorption, activated carbon, bisphenol S, bisphenol A

## Abstract

This study assessed three powdered activated carbons (BETM, COCO, and SIAL) commercialized in Brazil at the bench scale in agitated reactors, analyzing their kinetic behavior and adsorptive capacity for BPS and BPA in ultrapure water. BETM exhibited the highest adsorption capacities (Q^0^_max_) for BPS and BPA at 260.62 and 264.64 mg/g, respectively, followed by SIAL, with a Q^0^_max_ of 248.25 mg/g for BPS and for 231.20 mg/g BPA, and COCO, with a Q^0^_max_ of 136.51 mg/g for BPS and 150.03 mg/g for BPA. The Langmuir isotherm model can describe the processes well. A pseudo-second-order model can describe the adsorption kinetics, and SIAL carbon had the highest rate constants (7.45 × 10^−3^ mg/g/min for BPS and 2.84 × 10^−3^ mg/g/min for BPA). The Weber–Morris intraparticle diffusion model suggests intraparticle diffusion as the rate-limiting step of all adsorption processes. Boyd’s model confirmed more than the mechanism actuating in the bisphenol adsorption. The results suggest that adsorbents with basic surfaces, high specific surface areas, and high mesopore volumes tend to remove BPS and BPA efficiently. Therefore, activated carbons can effectively complement the existing treatment in Brazilian water treatment plants (WTPs).

## 1. Introduction

Intense technological development contributes to the development of new synthetic compounds widely used by the industrial sector and, consequently, by the population. Leachate from landfills and dumps, wastewater, fertilizers and pesticides, and oil derivatives can reach aquatic environments, contributing to the pollution of water bodies. Such compounds can contaminate surface and underground water bodies, adversely affecting aquatic biota. Moreover, their occurrence in water bodies used as supply sources poses risks to human health. Due to environmental and health concerns, health and environmental institutions worldwide are paying attention to these compounds and the possible effects associated with their occurrence in the environment.

Numerous studies have reported the occurrence of organic compounds in water bodies, known as contaminants of emerging concern. The United States Environmental Protection Agency (EPA) defines contaminants of emerging concern as compounds that do not have regulatory status. The EU has made it mandatory to monitor some compounds, while several are defined in the watch list [1]. Their environmental and human health effects are a threat and are not fully known. Moreover, such organic compounds occur in low concentrations in the environment, in the order of micrograms per liter (μg/L), nanograms per liter (ng/L), and even picograms per liter (pg/L).

Bisphenol A (BPA), bisphenol S (BPS), and bisphenol F (BPF) are considered emerging contaminants. These compounds can adversely affect ecosystems and human health, mainly because they are endocrine disruptors, i.e., such compounds interfere with the endocrine system of organisms, disrupting the production, release, transport, metabolism, binding, or action of hormones [2,3]. Depressive symptoms, especially in men, have been associated with exposure to BPA and BPS [4].

Bisphenols are constituents of various consumer goods, epoxy resins, water pipes, and thermal papers [5]. These versatile compounds can also occur in the environment due to the degradation of microplastics used to manufacture plastics [6]. As a result of its multiple applications, in Brazil, BPS has been detected in raw sewage, treated sewage, and human urine [7,8].

Concentrations of BPA have been detected in Lake Paranoá, located in the Federal District (FD), ranging from 0.002 to 1.231 μg/L [9,10,11,12]. In 1960, Brasília, the capital of Brazil, was founded, and since then, the population has increased. Population growth associated with the need for quality water was essential in triggering the 2017 water crisis, which resulted in an extensive rationing period. As an alternative solution, Lake Paranoá, the main water body in the FD, previously used as a receiving body, became a public water supply source with the installation of the Lago Norte Water Treatment Plant (WTP), producing 700 L/s of drinking water to supply nine administrative regions of the FD. This facility uses ultrafiltration technology to produce drinking water. However, ultrafiltration technology is ineffective at removing dissolved organic matter, which can include organic compounds such as bisphenols. Although the Lago Norte WTP can produce treated water to supply the region, other treatment steps must be implemented to ensure adequate removal of dissolved organic matter and other contaminants, thus ensuring the quality of the water supplied to the population. In this context, a challenge is posed to the WTP considering the drinking water quality and the consequent concern for public health.

The European Drinking Water Directive established a limit for BPA in drinking water of 2.5 µg/L [13]. Given the restrictions imposed, the use of BPS as the main substitute for BPA has been intensifying, which led to the production of 52.00 tons of this bisphenol in 2022, and is estimated to increase to 81.21 thousand tons in 2032 [14].

As more studies demonstrate the effects of BPS on human health and the environment [2,3], drinking water quality standards are becoming increasingly stringent from a legislative perspective. Therefore, it is necessary to assess economically viable water treatment technologies to remove this contaminant, and other bisphenols, diminishing the risks to human health associated with consuming contaminated water.

There are no Brazilian requirements for bisphenol levels in drinking water. However, due to health concerns, these compounds are expected to be included in future legislation revisions.

The adsorption of activated carbon is among the advanced technologies investigated for removing bisphenols with promising results [15]. Other methods applied to removing these organic pollutants, such as advanced oxidation processes, nanofiltration, and ion exchange resins, have some significant disadvantages compared with adsorption. Advanced oxidation processes, for example, can produce toxic by-products that require additional treatment. Ion exchange resins are effective at removing specific ions from water. However, BPS and BPA are predominantly in their non-ionized forms in aqueous media at pH levels below 8 and 9.6, respectively, which limits the ability of ion exchange resins to remove these compounds. The disadvantages of nanofiltration are associated with the initial installation cost and the system’s complexity, which can be significant. The cost of applying activated carbon is more advantageous than the technologies above since it does not require additional chemical reagents. In addition, it is possible to use existing units to apply activated carbon, such as rapid mixing units, which reduces installation costs. Another vantage of this technology is the possibility of producing activated carbons from various available materials [16,17,18,19]. Brazil, for example, has a very significant agroindustry, which makes it possible to use the waste generated to produce activated carbons with different properties that can be used in the water supply sector to remove different contaminants.

Furthermore, some studies indicate the capability of BPA’s removal by activated carbon to be over 90%, confirming the reusability [15].

In this context, it is crucial to evaluate the potential of activated carbons produced and marketed in Brazil to provide the water supply sector with information on the efficiency of these adsorbents. This will enable their application in water treatment with a focus on removing BPS.

Few studies have evaluated the removal of BPS, but no studies have analyzed the influence of the textural and chemical characteristics of activated carbons on the adsorption capacity of BPS. Therefore, the present study aimed to evaluate the removal of BPS in an ultrapure water matrix using three activated carbons commercialized in Brazil and produced from different raw materials, emphasizing the influence of their characteristics on the adsorption capacity.

This study also assessed the removal of BPA on the three carbons as a reference compound, as most studies on the occurrence of bisphenols have recorded the co-occurrence of both bisphenols in aquatic environments used as water supply sources.

## 2. Materials and Methods

### 2.1. Materials

BPS (98% purity) and BPA (99% purity) were purchased from Sigma Aldrich. Table 1 presents the physical and chemical properties of BPS and BPA.

Three activated carbons were chosen to assess BPS’s and BPA’s adsorbing potential: Two were granular carbons, acquired from national manufacturers; the first one was of vegetable origin, synthesized from coconut shell (COCO), and the second one was of mineral origin, synthesized from bitumen (BETM). The third activated carbon was a pulverized carbon of vegetable origin (SIAL), acquired from Sigma Aldrich.

### 2.2. Preparation of BPS and BPA Stock Solutions

To prepare the BPS and BPA stock solutions, 0.1 g of each bisphenol was weighed into two 250 mL Erlenmeyer flasks and 20 mL of ethyl alcohol (99.5%) was added to dissolve the bisphenols completely, followed by 200 mL of ultrapure water. The solution was heated on a hotplate for 60 min at a temperature between 75 and 80 °C to allow the alcohol to evaporate. Next, each solution was transferred to a 1 L volumetric flask, whose volume was filled with ultrapure water [25,26].

#### Study Waters

This study evaluated two study waters containing the target contaminants, BPS and BPA: ultrapure water + BPS (SW1) and ultrapure water + BPA (SW2). The ultrapure water was produced in the laboratory using a Milli-Q^®^ system (C79625, Merck Millipore, Darmstadt, Hesse, Germany). Aliquots of BPS and BPA stock solutions were added to the ultrapure water to obtain the study waters, SW1 and SW2, with 15 mg/L concentration each. After that, the pH of the study waters was adjusted to 7. This pH was chosen to reproduce the characteristics of water treated by ultrafiltration in the Lago Norte WTP.

It is important to mention that this study evaluated high concentrations of bisphenols, in the order of mg/L, since the UV-VIS spectrophotometry technique for quantifying these compounds is limited to low concentrations of these contaminants.

### 2.3. Quantification of BPS and BPA

The analyses to quantify bisphenol S and bisphenol A were performed by UV-VIS spectrophotometry (Hach DR 5000 UV-Vis Spectrophotometer) at wavelengths of 275 nm and 277 nm, respectively. Both wavelengths were chosen by scanning the stock solution (100 mg/L of each bisphenol) through wavelengths of 190 to 400 nm.

### 2.4. Characterization of Adsorbent Materials

The carbons COCO, BETM, and SIAL were characterized according to their physicochemical and textural properties, including specific surface area, point of zero charge (pH_PZC_), pH, and functional groups.

The carbons’ surface area and micropore and mesopore volumes were determined using a Quantachrome Nova Model 2200 surface area analyzer (Quantachrome Instruments, Boynton Beach, FL, USA). Varian 640-IR equipment (Varian Inc., Dresden, Germany) was used for FTIR analysis with a 600–4000 cm^−1^ scanning range.

Before the adsorption tests, the carbons were ground until 95% of the sample passed through a 325-mesh sieve with a 0.044 mm aperture. After grinding and sieving, the carbons were dried in an oven at 150 °C for 3 h and then stored in a desiccator to reach room temperature. Each carbon’s mass was weighed on an analytical balance and added to ultrapure CO_2_-free water to prepare the carbon suspensions. The suspensions were placed in a vacuum desiccator under a negative pressure of 600 mmHg to eliminate the air in the pores and hydrate the carbons [27].

### 2.5. Experimental Setup

Adsorption kinetics and equilibrium adsorption experiments were carried out on a bench scale using batch-activated reactors. The experiments were conducted in 500 mL Erlenmeyer flasks containing 300 mL of study water, at 25 °C. The experiments were performed under light conditions.

The experiments were carried out using the two study waters, SW1 and SW2, and the three activated carbons, BETM, COCO, and SIAL, resulting in a total of six combinations of experiments using the following combinations of study water and activated carbon:Ultrapure water + BPS + BETM.Ultrapure water + BPS + COCO.Ultrapure water + BPS + SIAL.Ultrapure water + BPA + BETM.Ultrapure water + BPA + COCO.Ultrapure water + BPA + SIAL.

### 2.6. Adsorption Kinetics Experiments

The experiments were conducted using a horizontal shaker (Fanem 2540 Shaker, São Paulo, Brazil). The study waters, SW1 and SW2, contained an initial concentration of approximately 15 mg/L of BPS and BPA in all the tests. The doses of activated carbon were 30 mg/L, 50 mg/L, and 30 mg/L, respectively, for BETM, COCO, and SIAL.

After adding the aliquot of each carbon suspension, the flasks were agitated for contact times of 0, 5, 10, 15, 30, 60, 120, 180, 360, 720, and 1440 min. After each contact time, treated water samples were taken from each flask and filtered through 0.22 µm cellulose ester membranes. Aliquots from filtered water were analyzed to quantify the remaining concentration of the target contaminant.

The experimental data were fit to the nonlinear pseudo-first-order, pseudo-second-order, intraparticle diffusion, and Boyd models. The tests were carried out in duplicate.

### 2.7. Adsorption Equilibrium Experiments

Table 2 presents the doses of each of the activated carbons adopted in this set of experiments. The contact time used during the adsorption equilibrium experiments was the equilibrium time obtained in the adsorption kinetic experiments for each pair of carbon and study water. The data were evaluated using mathematical models of isotherms (Langmuir and Freundlich) to determine each carbon’s BPS and BPA adsorptive capacities. The experiments were carried out in duplicate. Table 3 shows the equations and parameters of the isotherms and the kinetic models.

## 3. Results

### 3.1. Characterization of the Adsorbent Materials

Table 4 shows the chemical and textural characteristics of the three activated carbons evaluated in the present study. Bituminous carbon (BETM) and coconut shell carbon (COCO) are alkaline, while the SIAL carbon of vegetable origin is acidic. The pH values of the BETM and COCO carbons in the aqueous matrix were similar, with COCO carbons showing slightly higher alkalinity, with a pH of 10.0. In contrast, the reference carbon of SIAL showed a high acidity character, with a pH of 3.67.

Determining the point of zero charge (pH_PZC_) of adsorbents is essential to understanding the electrical characteristics of the surface of these materials and inferring the adsorption capacity of the adsorbate in the adsorption process. When the charges of the adsorbate and the adsorbent coincide, there is a trend for electrostatic repulsion. The pH_PZC_ represents the pH value at which the surface of the adsorbent has a neutral charge. When the pH of the medium is higher than the pH_PZC_, the surface charge of the adsorbent becomes negative, while pH values lower than the pH_PZC_ result in a positive adsorbent surface charge.

The adsorption process under pH values can promote electrostatic interactions between the bisphenols and the activated carbons. However, regarding the pKa values of BPS and BPA, 8 and 9.6, respectively, it is possible to infer that in an aqueous matrix with a pH of approximately 7.0, BPS and BPA will predominantly be in their non-ionized forms.

The specific surface area of adsorbents influences the removal efficiency of micropollutants, as this property is associated with the ability to fill or utilize the active sites available on the surface area of the adsorbent. The volume of micropores and mesopores, in turn, not only provides information on the structure of the activated carbon but also indicates the trend of a compound to access the pores of the adsorbent material as a function of its molecular size. Figure 1 displays the N_2_ adsorption/desorption isotherms obtained during the textural analysis of the three activated carbons.

The adsorption isotherm of the carbon SIAL can be interpreted as a combination of type I(b) and IV isotherms, according to the IUPAC classification, common in microporous and mesoporous adsorbents, respectively [28]. Type I(b) isotherms are characterized by high N_2_ adsorption rates at low relative pressures (p/po < 0.05) due to the filling of narrow micropores (ultramicropores) as a consequence of the overlapping of the adjacent pore walls’ adsorption potentials, which are very close. In addition, the presence of a wide knee in the curve results from the filling of larger micropores (supermicropores). Type IV isotherms show a continuous increase in adsorption until near saturation, initially due to the formation of multiple adsorbed layers and, later, at higher pressures, due to the filling of mesopores by capillarity [29].

The hysteresis loop in the isotherm SIAL activated carbon at higher relative pressures, above p/po = 0.6, in which the N2 amount adsorbed during desorption is greater than during adsorption, indicates the mesopores’ presence on its surface. According to the IUPAC classification, the observed hysteresis loop has intermediate characteristics between the H3 and H4 patterns, typical of adsorbents with slit pores [28]. On the other hand, the isotherms of BETM and COCO activated carbons show characteristics more similar to type I(b), suggesting a predominance of ultramicropores and supermicropores on their structures.

SIAL carbon has a mesopore volume 82% higher than BETM carbon and 93% higher than COCO carbon. Regarding specific surface area, SIAL carbon has a specific surface area 37% higher than BETM carbon and 52% higher than COCO carbon, corresponding to a micropore volume higher than BETM and COCO carbons. Considering the molecular diameters of BPS (0.625 nm) and BPA (1.068 nm), shown in Table 1, it is expected that more mesoporous carbons will have greater efficiency in removing both compounds, given that the mesopores are between 2 and 50 nm in size.

The three adsorbents differed in their surface characteristics, both in surface area and in micropore and mesopore volume, where, according to Table 4, the specific surface area, mesopore volume, and micropore volume of SIAL (1397 m^2^/g; 0.423 cm^3^/g; 0.623 cm^3^/g) are greater than those of BETM (880 m^2^/g; 0.076 cm^3^/g; 0.397 cm^3^/g), which in turn are greater than those of COCO (673 m^2^/g; 0.031 cm^3^/g; 0.314 cm^3^/g). Regarding the surface characteristics, the SIAL carbon was expected to show the higher capacity of bisphenol removal.

Figure 2 illustrates the infrared spectra of each carbon. In general, the most intense absorptions were observed only in the SIAL carbon, with lower intensities in the other samples.

In the spectrum of SIAL carbon, a strong absorption stretch at 1631 cm^−1^ (C=O) and a broad absorption stretch at 3417 cm^−1^ (O-H) indicate the presence of the carboxylic alcohol function. Carboxylic functional groups, phenols, lactones, and alcohols intensify charcoal’s hydrophilic character, lowering the pH [29]. A weak-to-medium absorption stretch at 2360 cm^−1^ indicates the presence of the thiol group (S-H). Thiol groups have a pKa between 9 and 10, which intensifies the acidic nature of this adsorbent [30]. The absorptions at 1216 cm^−1^ (C-O), 3417 cm^−1^ (O-H), and 1075 cm^−1^ are compatible with the primary alcohol function.

For the BETM and COCO carbons, there is no absorption between 1820 and 1630 cm^−1^, ruling out the possibility of any carbonyl function (C=O). In the spectrum of BETM carbon, there is a strong stretching absorption at 1080 cm^−1^ (C-O), indicating the presence of the ether function. Ether groups, pyrones, and chromenes contribute to the basic surface properties [31].

The highest intensity absorptions were only significantly observed in SIAL carbon, with lower intensity in the other carbons. It is suggested that the presence of the functional groups reported is greater on the surface of SIAL carbon than BETM and COCO carbon, especially in the spectrum of COCO carbon, which showed practically no detectable stretching.

### 3.2. Adsorption Kinetics Experiments

Figure 3 displays the average values of each bisphenol’s (C) remaining concentration relative to its initial concentration (C_0_) over time for each carbon used (BETM, COCO, and SIAL). Equilibrium of adsorption was achieved for all types of carbons within 120 min (2 h). Thus, this time interval was established as the contact time for the adsorption equilibrium tests.

Nonlinear forms of the models provide better fits for the pseudo-first-order and pseudo-second-order kinetic models than linear forms, thus avoiding distortions in the kinetic parameters [32].

The typical behavior of both bisphenols’ remaining fraction was observed for all the carbons. In the first few minutes of contact (30 min), the activated carbons removed over 40% of BPS and BPA, and for all the carbons the adsorption equilibrium was reached in 120 min (2 h). This time was, therefore, defined as the equilibrium time to be adopted in the equilibrium adsorption experiments to obtain the adsorption isotherms.

Table 5 depicts the results obtained after fitting the experimental data to the pseudo-first-order, pseudo-second-order, intraparticle diffusion Weber–Morris, and Boyd kinetic model. Figure 4 illustrates the experimental data of the BPS and BPA kinetic adsorption experiments using each activated carbon (BETM, COCO, and SIAL) and their respective fittings to the models.

The pseudo-first-order model’s coefficient of determination (R^2^) for the BPS and BPA adsorption onto the three carbons was relatively high. However, there were differences between the experimental and calculated equilibrium adsorption capacities (q_e_), as evidenced by the standard deviation associated with each q_e_ value and the high SQR values. Conversely, the experimental data exhibited good pseudo-second-order model fitting, with R^2^ ≥ 0.99 for almost all carbons and lower SQR values. The equilibrium adsorption capacities estimated by the pseudo-second-order model also present lower standard deviations, approaching the experimental equilibrium adsorption capacities. Therefore, the experimental data were better fit by the pseudo-second-order model. Since this kinetic model assumes that chemisorption predominates in the process [33], the surface interactions between the adsorbate and adsorbent likely involve electron sharing or exchange.

In recent studies involving BPS adsorption on activated carbons in ultrapure water matrices, the experimental data have appropriately fit the pseudo-second-order kinetic model as shown in Table 6 [23,34,35,36,37,38,39,40,41]. Nevertheless, none of the studies obtained better experimental data fits with the pseudo-first-order kinetic model. Thus, the present study confirms the findings in the literature. It is worth noting that applying kinetic models is important to predict the rate at which the contaminant is removed from aqueous matrices, enabling the design of appropriate full-scale units to perform the adsorption process.

Table 6 also shows that the rate constant attributed to the SIAL carbon showed the highest value among the evaluated carbons, indicating the highest rate of the BPS adsorption process.

Figure 5 and Figure 6 display the experimental data from BPS and BPA adsorption experiments on each activated carbon (BETM, COCO, and SIAL) and their respective fits to the Weber–Morris and Boyd intraparticle diffusion models.

Several studies have applied the intraparticle diffusion model to verify the limiting step in the kinetic adsorption process [23,36,37,38]. Solute adsorption in an aqueous medium involves adsorbate mass transfer (film diffusion), surface diffusion, and pore diffusion (intraparticle diffusion) [42]. Film diffusion occurs when the adsorbate diffuses from the aqueous medium to the external surface of the solid adsorbent and the fluid layer favors the adsorbate retention. Intraparticle diffusion occurs when the adsorbate diffuses into the pores. Finally, in the surface diffusion step the adsorbate is adhered to the pores’ surfaces, resulting in adsorption [43].

The linear fitting of the experimental data of BPS and BPA adsorption onto the three carbons in the intraparticle diffusion model did not intercept the origin of the graphic, suggesting the intraparticle diffusion is not the sole limiting step in the adsorption process and that multiple stages likely limit both bisphenols’ adsorption kinetics. During the initial stage, both bisphenols rapidly migrate from the water to the external surface of the adsorbents, characterized by film diffusion. The second phase encompasses intraparticle or pore diffusion, resulting in a gradual increase in adsorption capacity. In the third phase, the pores of the adsorbent reach saturation, establishing the adsorption equilibrium [44]. A behavior similar to that found herein was observed in studies which investigated BPS adsorption on activated carbons [23,36,37,38].

Regarding the rate constant of the intraparticle diffusion model for each stage (k_p_), k_p1_ exhibited a greater value than k_p2_ for all six adsorptive processes. Thus, the first adsorption stage, characterized by film diffusion, occurred over a shorter time. Therefore, intraparticle diffusion and film diffusion were the main mechanisms responsible for the rate-limiting step of the adsorption process of both contaminants, influencing the overall kinetics of bisphenol adsorption on the three activated carbons.

According to Kajjumba et al. (2018), if the system is characterized by poor mixing and low adsorbate concentration, film diffusion becomes the rate-controlling step; otherwise, intraparticle diffusion controls the process [42]. In this study, the concentrations of bisphenols used were high, which probably facilitated the diffusion of contaminants through the film. This led to intraparticle diffusion to control the adsorption of bisphenols on the three carbons. Therefore, intraparticle diffusion controlled the adsorption of BPS and BPA onto the three activated carbons [42].

For BPA and BPS, in the first stage of the adsorption, SIAL carbon showed lower resistance to film diffusion compared to BETM and COCO carbons, as indicated by their higher intraparticle diffusion rate constants. Notably, in the second stage, for BPA, SIAL carbon presented a k_p2_ value approximately three times higher than for BPS, indicating slower adsorption of BPS. In contrast, for BPS, BETM carbon showed a k_p2_ value approximately twice as high as for BPA, suggesting slower adsorption of BPS. COCO carbon showed comparable k_p1_ and k_p2_ values for both bisphenols.

The parameter C in the intraparticle diffusion model equation represents the thickness of the boundary layer. The values of C determine the effect of the boundary layer on the adsorption process; the higher the values, the higher the effect of such a boundary [42]. Accordingly, increasing values of C indicate that the thickness of the boundary layer has also increased proportionally, causing the process to reach equilibrium.

It is worth noting that the categorization of kinetic data into specific groups is a challenging task since there are no established criteria for this division [45]. Therefore, exploring various approaches to dividing kinetic data is beneficial in finding the one that offers maximum efficiency with ideal statistical parameters. Thus, in this study, the best approach yielded higher R^2^ values in the first and second stages of the adsorption mechanism.

The Boyd model was applied to investigate the process’s limiting step, either intraparticle diffusion or diffusion through the liquid film, in the adsorption of BPS and BPA on the studied activated carbons. The linear fittings to the Boyd model observed in Figure 6 and the coefficients of determination of each line fitting listed in Table 6 indicate that the linear regressions do not intercept the origin, which means that the diffusion through the film or other external mass transport mechanisms may be present in the adsorption processes of bisphenols. This behavior confirms the results obtained from applying the intraparticle diffusion model, as both models indicate that other mechanisms, besides intraparticle diffusion, occur in the adsorption process of BPS and BPA on the three carbons, such as diffusion through the liquid film.

Based on the Weber–Morris and Boyd intraparticle diffusion models, it is possible to verify that the adsorption of BPS and BPA on COCO, BETM, and SIAL carbons involves diffusion mechanisms through the liquid film and intraparticle diffusion and confirms the findings reported in a recent study conducted by Aziz et al. (2022), who assessed BPS adsorption onto activated carbon (RPAC) synthesized from rambutan bark or Indonesian lychee; the adsorbent had a specific surface area of 402.68 m^2^/g. The concentrations of BPS varied between 5 and 30 mg/L, and the dose of carbon adopted was equal to 1 g/L, which was considerably higher than those adopted for the activated carbons studied herein [37].

When subjected to each activated carbon, BPS and BPA revealed similar kinetic behaviors, suggesting that mechanisms and interactions involved in BPS and BPA transport and retention on adsorbent materials are similar due, likely, to their structural similarity. Wang et al. (2021) produced an adsorbent material using chicken feathers and heteroatom sources (nitrogen and sulfur) as precursors, using zinc chloride and basic magnesium carbonate as double activation agents (MZ-NSPC) [41]. The adsorbent had a specific surface area of 1383.5 m^2^/g and a micropore volume of 0.36 cm^3^/g. MZ-NSPC was investigated as to its BPS and BPF adsorption capacity. The initial concentration of each bisphenol adopted in the adsorption experiments was 110 mg/L, and the dose of the adsorbent was 100 mg/L.

### 3.3. Adsorption Equilibrium Experiments

Figure 7 displays the removal percentages of BPS and BPA onto each carbon, respectively, as a function of the carbon dosage at the equilibrium condition using a contact time of 120 min.

Activated carbon of mineral origin (BETM) displayed the highest BPS and BPA removal percentages in the entire range of analyzed dosages, followed by SIAL and COCO carbons. BPS adsorption on all the carbons was higher than that of BPA. BPA removal ranged from 33% to 93% in the dosage range applied to bituminous carbon, while BPS removal ranged from 36% to 98%. COCO carbon was the least favorable for removing bisphenols, probably due to its physicochemical characteristics, such as a lower specific surface area and mesopore volume lower than those of BETM and SIAL carbons. Considering the carbons’ surface characteristics, it was expected that the SIAL carbon would show a higher percentage of removal than the BETM and COCO carbons. However, the physicochemical characteristics of BETM carbon probably favored the adsorption of both bisphenols.

Concerning the pKa values of BPS and BPA, 8 and 9.6, respectively, and knowing that the pKa of a chemical substance represents the pH value of the medium in which the substance occurs, in equilibrium, in its non-ionized and ionized forms, it is likely that in the ultrapure water with a pH of about 7.0, BPS and BPA are predominantly in their non-ionized forms.

The point of zero charge (pH_PZC_) represents the pH value at which the surface of the adsorbent has a neutral charge. If the pH of the medium is higher than the pH_PZC_, the surface charge of the adsorbent becomes negative, while at pH values lower than the pH_PZC_ the adsorbents’ surface presents a positive charge. According to the pH_PZC_ obtained for each adsorbent, at pH = 7, BETM (pH_PZC_ = 9.06) and COCO (pH_PZC_ = 9.97) carbons acquire a positive surface charge. Conversely, SIAL carbon (pH_PZC_ = 3.08) acquires a negative surface charge under the same pH conditions. Therefore, the electrostatic charges on the three carbons’ surface likely slightly influence the adsorptive process; thus, there is a low probability that the electrostatic interactions between bisphenols and activated carbons predominate in the adsorption process.

However, the Log K_OW_ of BPS and BPA, 1.2 and 3.32, respectively, indicate the hydrophobic nature of both bisphenols in neutral pH conditions.

Regarding activated carbon’s acidic–basic character, the pH value of BETM carbon is highly alkaline (pH = 9.6), suggesting the presence of basic functional groups on its surface. In contrast, SIAL carbon is strongly acidic (pH = 3.7). The higher removal efficiency of BPS and BPA on BETM carbon may be associated with the basic functional groups on the carbon’s surface, which intensify its hydrophobic character and, in turn, promote the hydrophobic interactions between the bisphenols and this carbon.

Thus, the interactions between hydrophobic groups on the carbons’ surfaces and bisphenols may have been determining factors in the adsorption process.

The experimental data of BPS and BPA adsorption obtained at the equilibrium condition were analyzed for their fit to the Langmuir and Freundlich isotherm models. Table 7 shows the parameter values for each isotherm model. Figure 8 shows the adsorption isotherms, obtained at 25 °C, for BPS and BPA on the three activated carbons.

The experimental data from each BPS and BPA adsorption assay better fit the Langmuir isotherm model for the three activated carbons. According to the Langmuir model’s assumptions, the adsorption of both bisphenols occurs in a monolayer. Additionally, there is a defined number of active sites, each retaining one molecule of bisphenol. The active sites have the same adsorption energy, and the adsorbed molecules do not interact with each other.

The separation factor (R_L_) in the Langmuir isotherm model indicates the feasibility of the adsorption process, whether the adsorption is favorable (R_L_ < 1), irreversible (R_L_ = 0), linear (R_L_ = 1), or unfavorable (R_L_ > 1). It can be calculated from the following equation:(1)$$R_{L}=\frac{1}{1+k_{L}C_{0}}$$
where *k_L_* (L/mg) is the Langmuir constant and *C*_0_ (mg/L) is the initial concentration of the adsorbate [46].

The values of *R_L_* reveal that the adsorption of BPS and BPA on the three adsorbents is favorable.

The Langmuir isotherm exhibited higher determination coefficients (R^2^) and lower SQR values than the Freundlich model, indicating that such isotherms better represent the adsorption process of BPS and BPA on BETM, COCO, and SIAL carbons.

The Langmuir model shows that the BETM carbon displayed higher adsorption capacities (Q^0^_max_) of BPS and BPA, 260.62 and 264.64 mg/g, respectively. It was followed by the SIAL carbon, with a Q^0^_max_ value of 248.25 mg/g for BPS and 231.20 mg/g for BPA. Finally, the COCO carbon presented the lowest Q^0^_max_ among the studied adsorbents, with 136.51 mg/g for BPS and 150.03 mg/g for BPA.

The results obtained in the present study are consistent with recent studies concerning BPS adsorption on activated carbons [23,34,35,36,37,38,39], as the experimental data adequately fit the Langmuir model. Nevertheless, this study obtained more satisfactory results the studies above.

Aziz et al. (2022) investigated the adsorption of BPS in ultrapure water onto an activated carbon with a specific surface area lower than those of three carbons evaluated herein (402.68 m2/g) [37]. In the adsorption tests, the carbon dosage applied was 1 g/L, and the initial concentration of BPS varied in the 5–30 mg/L range. The maximum BPS adsorption capacity, 27.89 mg/g, was lower than that found in the present study. Zhao et al. (2022) achieved a maximum BPS adsorption capacity of 83.19 mg/g, applying 250 mg/L of commercial activated carbon in adsorption tests using ultrapure water; the initial concentration of BPS varied in the 20–50 mg/L range [39].

Unlike the present work, in the study by Zhang et al. (2020), the activated carbon’s maximum BPS adsorption capacity (617.29 mg/g) was higher than that found herein [23]. However, despite the carbon having a BET surface area of 722.22 m^2^/g, the carbon dosage range applied was considerably high (200–600 mg/L) for a range of BPS concentrations in the ultrapure water of 10 to 90 mg/L [23]. In contrast, in this study, the dosage ranges of carbons varied between 10 and 80 mg/L for a BPS and BPA concentration of 15 mg/L in the ultrapure water.

Table 8 shows the results of the studies evaluating BPS and BPA adsorptions after fitting the experimental data to the Langmuir isotherm model.

BETM carbon demonstrated the highest BPS and BPA adsorption capacity among the studied carbons. This carbon presents a specific surface area higher than the activated carbon from coconut (COCO) and lower than the SIAL carbon of vegetal origin. Comparing the BETM and COCO carbons with basic characteristics, the higher adsorption capacity of BPS and BPA onto BETM carbon might be associated with its elevated specific surface area and higher volume of mesopores, which provide a higher number of accessible adsorption sites. Comparing BETM and SIAL carbons, as previously mentioned, despite SIAL carbon presenting a higher specific surface area and larger volume of mesopores, the presence of acidic functional groups on the surface probably reduced the surface’s hydrophobicity, affecting the hydrophobic interactions with bisphenols in the adsorption process.

It is also noteworthy that BPS and BPA exhibited similar behaviors in the adsorption process onto the three activated carbons, as evidenced by the maximum adsorption capacity values obtained from the Langmuir isotherm models. Such evidence corroborates the assumption that the similar adsorption behaviors of both BPA and BPS on adsorbent materials result from the structural similarity between both bisphenols. In a study conducted by Wang et al. (2021) [41], the authors observed that the removal capacity of BPS and BPF in functionalized activated carbons was similar, 295.8 mg/g and 308.7 mg/g, respectively. The experimental data also fit the Langmuir isotherm satisfactorily.

For future research, it is essential to evaluate the simultaneous adsorption of BPS and BPA on the activated carbons studied herein, as contaminants often co-occur in aquatic environments used as water supplies. The simultaneous adsorption of both contaminants in real surface water matrices, including Paranoá ultrafiltered water, should also be investigated using concentrations of both bisphenols in the order of µg/L, lower than those assessed in the present study. Carrying out experiments in conditions closer to reality will provide a more complete understanding of the behavior of activated carbons in removing bisphenols in real aqueous matrices. Further research, especially at the Lago Norte WTP, will make it possible to complement the treatment of water supplied to the population of the Federal District. These issues are essential to developing efficient technologies based on the adsorption process for bisphenol removal in full-scale water treatment systems, promoting significant advances in treating water with a focus on removing contaminants of emerging concern and producing chemically safe drinking water. Finally, experiments are essential to evaluate the feasibility of reusing carbons to improve the economic viability of the adsorption process.

## 4. Conclusions

The textural characterization of the three activated carbons studied indicated that the SIAL carbon presents a more abundant mesoporous structure and the highest BET surface area (1397 m^2^/g). In contrast, the BETM and COCO carbons present lower BET surface areas, 880 m^2^/g and 673 m^2^/g, respectively, with a predominantly microporous structure.

BETM carbon exhibited better performance in all kinetic and adsorption equilibrium assays. Hydrophobic interactions between the carbons’ surfaces and BPS and BPA were probably determining factors in adsorption.

Fittings to the kinetic models indicated that the pseudo-second-order model best represented the experimental data. Fittings of the experimental data to the intraparticle diffusion and Boyd models suggest that the adsorption process may occur by two different diffusive mechanisms, intraparticle diffusion and film diffusion. Fittings of the experimental data to the intraparticle diffusion model suggest that intraparticle diffusion is the main mechanism responsible for the rate-limiting step of the adsorption process of BPS and BPA.

The experimental data of BPS and BPA adsorption assays in the equilibrium state better fit the Langmuir isotherm model for the three activated carbons studied. The Langmuir model shows that BETM carbon exhibited higher adsorption capacities (Q^0^_max_) for BPS and BPA of 260.62 and 264.64 mg/g, respectively. It was followed by SIAL carbon, with Q^0^_max_ values of 248.25 mg/g for BPS and 231.20 mg/g for BPA. Finally, COCO carbon presented Q^0^_max_ values of 136.51 mg/g for BPS and 150.03 mg/g for BPA.

It is important to highlight that BPS and BPA exhibited analogous behavioral patterns in both the kinetic and adsorption processes onto the activated carbons in equilibrium conditions. This evidence reinforces the premise that BPS’s and BPA’s structural similarity may indicate equivalent adsorption behaviors of both contaminants on adsorbent materials.

The results of this study suggest that adsorbents with basic surface characteristics, a high specific surface area, and a high mesopore volume favor bisphenol adsorption processes in an aqueous medium.

## Figures and Tables

**Figure 1 ijerph-21-00792-f001:**
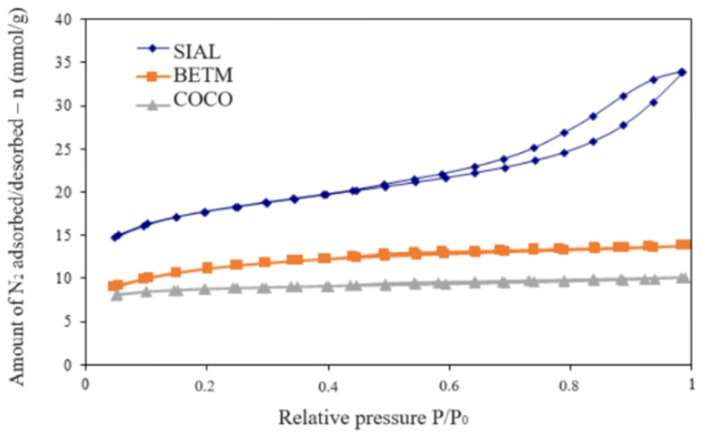
N_2_ adsorption/desorption isotherms for the three activated carbons.

**Figure 2 ijerph-21-00792-f002:**
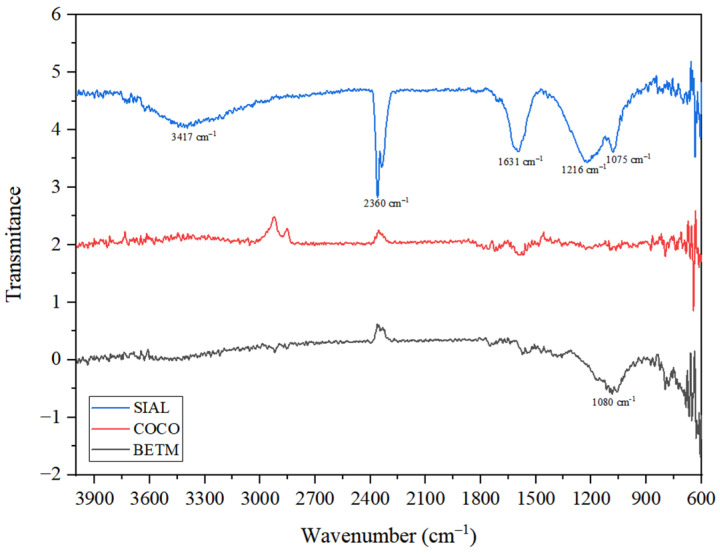
Infrared spectra obtained for each activated carbon through FTIR analysis.

**Figure 3 ijerph-21-00792-f003:**
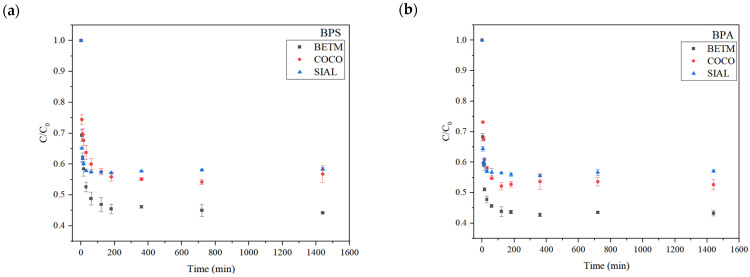
Bisphenol S’s (**a**) and bisphenol A’s (**b**) remaining fraction (C/C_0_) as a function of contact time, representing the average values of duplicate trials. C_0_: initial concentration of BPS (15 mg/L). C: BPS concentration in water after the contact time. Carbon dosage applied: BETM (30 mg/L), COCO (50 mg/L), and SIAL (30 mg/L). Aqueous matrix: ultrapure water. pH = 7. Temperature = 25 °C. Bars represent the standard deviations of the experimental C/C_0_ average values.

**Figure 4 ijerph-21-00792-f004:**
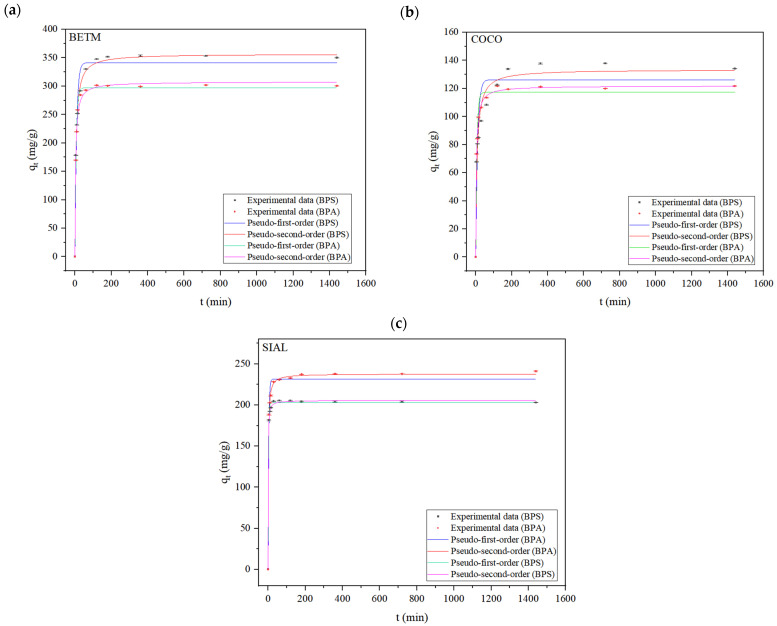
Experimental kinetic data adjusted to the pseudo-first-order and pseudo-second-order models for BPS and BPA adsorption on BETM carbon (**a**), COCO carbon (**b**), and SIAL carbon (**c**), representing the average values of duplicate trials. Carbon dosage applied: BETM (30 mg/L), COCO (50 mg/L), SIAL (30 mg/L); BPS dosage applied: 15 mg/L; BPA dosage applied: 15 mg/L. Aqueous matrix: ultrapure water. pH = 7. Temperature = 25 °C. Bars represent the standard deviations of the experimental q_t_ average values.

**Figure 5 ijerph-21-00792-f005:**
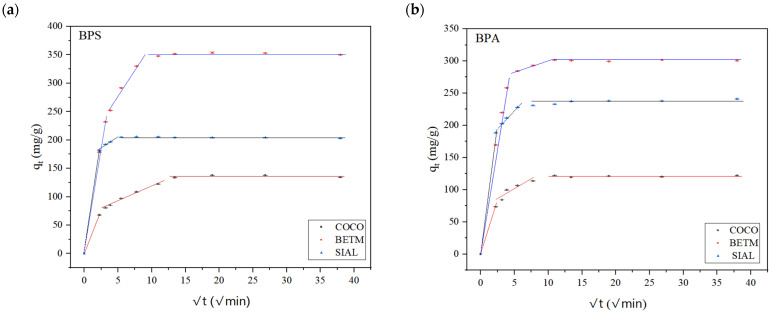
Experimental kinetic data fitting to the Weber–Morris intraparticle diffusion model for BPS (**a**) and BPA (**b**) adsorption on COCO, BETM, and SIAL carbons. Carbon dosage applied: BETM (30 mg/L), COCO (50 mg/L), and SIAL (30 mg/L); BPS initial concentration: 15 mg/L. Aqueous matrix: ultrapure water. pH = 7. Temperature = 25 °C. Bars represent the standard deviations of the experimental q_t_ average values.

**Figure 6 ijerph-21-00792-f006:**
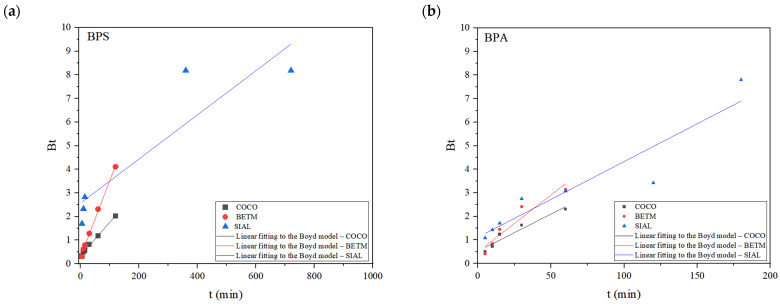
Experimental kinetic data adjusted to the Boyd model for BPS (**a**) and BPA (**b**) adsorption on COCO, BETM, and SIAL carbons. Carbon dosage applied: BETM (30 mg/L), COCO (50 mg/L), and SIAL (30 mg/L); BPS dosage applied: 15 mg/L; BPS dosage applied: 15 mg/L. Aqueous matrix: ultrapure water. pH = 7. Temperature = 25 °C.

**Figure 7 ijerph-21-00792-f007:**
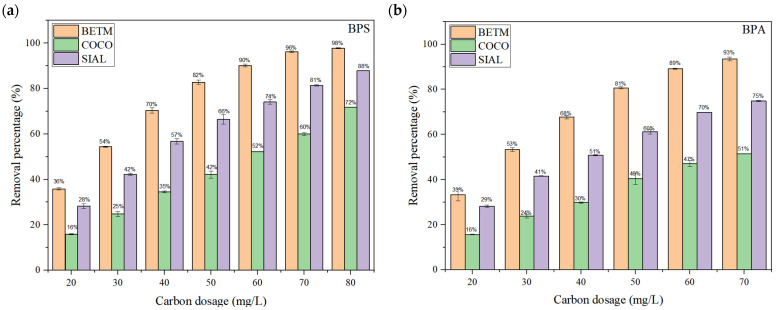
Percentage of BPS removal at different dosages of activated carbon for an initial BPS concentration of 15 mg/L (**a**) and an initial BPA concentration of 15 mg/L (**b**) at the equilibrium condition. pH = 7. Temperature = 25 °C. Contact time = 120 min. Bars represent the standard deviations of the experimental average values.

**Figure 8 ijerph-21-00792-f008:**
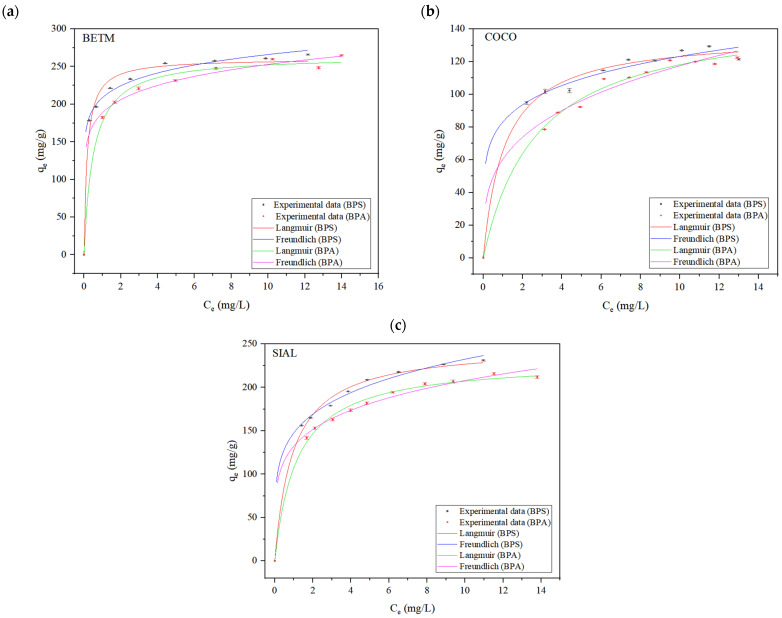
Adsorption isotherms of BETM carbon (**a**), COCO carbon (**b**), and SIAL carbon (**c**) for BPS and BPA. q_e_: adsorption capacity under equilibrium condition (mg/g), C_e_: concentration of adsorbate in medium at equilibrium state (mg/L). BPS (15 mg/L) and BPA (15 mg/L). pH = 7. Contact time = 120 min. Temperature = 25 °C.

**Table 1 ijerph-21-00792-t001:** Physical and chemical properties of BPS and BPA [20,21,22,23,24].

Molecular Formula	Molar Mass (g/mol)	Molecular Diameter (nm)	Log Kow	pKa	Water Solubility (mg/L)
C_12_H_10_O_4_S	250.27	0.625	1.2	8	715
C_15_H_16_O2	228.29	1.068	3.32	9.6	300

**Table 2 ijerph-21-00792-t002:** Doses of each activated carbon (mg/L) used in the equilibrium adsorption experiments.

Activated Carbon	SW1 (15 mg/L of BPS)	SW2 (15 mg/L of BPS)
BETM	10; 20; 30; 40; 50; 70; and 80	5; 10; 20; 30; 40; 50; and 70
COCO	20; 30; 40; 50; 70; 80; 90; and 100	20; 30; 40; 50; 70; 80; 90; 100; and 110
SIAL	20; 30; 40; 50; 70; 80; and 90	10; 20; 30; 40; 50; 70; 80; 90; and 100

**Table 3 ijerph-21-00792-t003:** Equations and parameters of isotherms and kinetics models.

Model	Name	Nonlinear Equation	Parameters
**Isotherm**	Langmuir	$q_{e}=\frac{Q_{max}^{0}K_{L}C_{e}}{1+K_{L}C_{e}}$	*C_e_*: Concentration of bisphenol at equilibrium (mg/L) *q_e_*: Amount of adsorbate adsorbed per unit mass of adsorbent at equilibrium (mg/g) $Q_{max}^{0}$: Theoretical maximum adsorption capacity (mg/g) *K_L_*: Langmuir constant related to the rate of adsorption *n* and *K_F_*: Freundlich constants
	Freundlich	$q_{e}=K_{F}C_{e}^{n}$	
**Kinetics**	Pseudo-first-order	$q_{t}=q_{e}(1-e^{-k_{1}t})$	*q_t_*: Amount of adsorbate adsorbed per unit mass of adsorbent at time t constant (mg/g) *k*_1_: Pseudo-first-order rate constant *k*_2_: Second-order rate constant *k_p_*: Intraparticle diffusion rate constant *C*: Intraparticle diffusion rate constant *F*: Fraction between *q_t_* and *q_e_* *Bt*: Mathematical function of *F*
	Pseudo-second-order	$q_{t}=\frac{{k_{2}q}_{e}^{2}t}{1+k_{2}q_{e}}$	
	Intraparticle diffusion	$q_{t}=k_{p}\sqrt{t}+C$	
	Boyd	$F=1-\left(\frac{6}{\pi^{2}}\right)\sum_{n=1}^{\infty }\left(\frac{1}{\pi^{2}}\right)e^{-\pi^{2}Bt}$	

**Table 4 ijerph-21-00792-t004:** Chemical and textural characteristics of the activated carbons.

Activated Carbon	Surface Area BET (m^2^/g)	V_0.95_ (cm^3^/g)	Micropore Volume (cm^3^/g)	Mesopore Volume (cm^3^/g)	pH	pH_PZC_
BETM	880	0.473	0.397	0.076	9.60	9.06
COCO	673	0.345	0.314	0.031	10.00	9.97
SIAL	1397	1.054	0.632	0.423	3.67	3.08

**Table 5 ijerph-21-00792-t005:** Kinetic parameters obtained after fitting the experimental data to the pseudo-first-order, pseudo-second-order, intraparticle diffusion Weber–Morris, and Boyd kinetic models.

		BPS			BPA		
Kinetic Model	Parameter	BETM	COCO	SIAL	BETM	COCO	SIAL
**Pseudo-first-order (PFO)**	q_e_ (exp) (mg/g)	351.23	133.24	204.21	300.80	120.84	237.41
	q_e_ (calc) (mg/g)	341.42 ± 8.00	133.62 ± 5.01	202.80 ± 1.27	297.12 ± 2.87	117.34 ± 2.47	231.63 ± 3.54
	k_1_ (min^−1^)	0.1102 ± 1.25 × 10^−2^	0.1119 ± 2.04 × 10^−2^	0.4314 ± 3.30 × 10^−2^	0.1474 ± 7.6 × 10^−3^	0.1499 ± 1.6 × 10^−4^	0.2927 ± 3.65 × 10^−2^
	R^2^	0.9685	0.9205	0.9967	0.9940	0.9713	0.9816
	SQR	37.71	14.85	1.24	5.15	3.82	8.97
**Pseudo-second-order (PSO)**	q_e_ (calc) (mg/g)	356.80 ± 2.93	140.70 ± 3.03	205.57 ± 0.71	307.51 ± 2.96	121.84 ± 1.09	237.84 ± 1.33
	k_2_ (mg/g/min)	5.11 × 10^−4^ ± 3.00 × 10^−5^	1.24 × 10^−3^ ± 1.87 × 10^−4^	7.45 × 10^−3^ ± 6.42 × 10^−4^	8.86 × 10^−4^ ± 7.10 × 10^−5^	2.22 × 10^−3^ ± 1.65 × 10^−4^	2.84 × 10^−3^ ± 2.17 × 10^−4^
	R^2^	0.9968	0.9785	0.9992	0.9951	0.9958	0.9981
	SQR	0.42	0.45	0.28	0.41	0.56	0.95
**Intraparticle** **Diffusion Weber–Morris**	C_1_	2.71 ± 11.32	2.00 ± 8.37	0.00	5.80 ± 11.17	4.42 ± 8.76	0.00
	$kp_{1}\;(\mathrm{mg}/g/\sqrt{min})$	74.56 ± 5.06	26.35 ± 3.74	81.13	67.33 ± 4.08	25.83 ± 3.20	84.28
	R^2^_1_	0.9954	0.9802	1	0.9927	0.9702	1
	C_2_	197.39 ± 17.57	66.30 ± 1.99	177.71 ± 5.84	229.16 ± 20.92	88.60 ± 1.86	163.15 ± 3.34
	$kp_{2}\;(\mathrm{mg}/g/\sqrt{min})$	14.94 ± 2.56	5.14 ± 0.24	4.09 ± 1.19	8.65 ± 3.54	3.09 ± 0.25	12.08 ± 0.86
	R^2^_2_	0.9189	0.9914	0.7971	0.8568	0.9873	0.9899
	C_3_	350.56	136.36 ± 3.86	205.86 ± 0.35	296.78 ± 2.84	120.36 ± 1.28	229.02 ± 1.74
	$kp_{3}\;(\mathrm{mg}/g/\sqrt{min})$	0.03 ± 0.13	−0.02 ± 0.15	−0.08 ± 0.02	0.14 ± 0.13	0.02 ± 0.05	0.35 ± 0.09
	R^2^_3_	0.0213	0.0075	0.8436	0.2259	0.0527	0.7752
**Boyd**	Angular coefficient (B)	0.03 ± 9.0 × 10^−4^	0.01 ± 6.0 × 10^−4^	0.01 ± 2.5 × 10^−3^	0.05 ± 8.9 × 10^−3^	0.03 ± 4.9 × 10^−3^	0.03 ± 5.3 × 10^−3^
	Linear coefficient	0.28 ± 4.9 × 10^−2^	0.33 ± 3.3 × 10^−2^	2.57 ± 0.89	0.49 ± 0.28	0.52 ± 0.15	1.11 ± 0.46
	R^2^	0.9972	0.9936	0.8285	0.9075	0.9312	0.8805

q_e_ (exp): experimental adsorbed amount of adsorbate per unit mass of adsorbent at equilibrium; q_e_ (calc): calculated adsorbed amount of adsorbate per unit mass of adsorbent at equilibrium; R^2^: coefficient of determination; k_1_: rate constant of PFO equation; k_2_: rate constant of PSO equation; SQR: sum of squares of residuals. k_p1_: intraparticle diffusion rate constant of stage 1; C_1_: intraparticle diffusion constant of stage 1; R^2^_1_: determination coefficient of stage 1; k_p2_: intraparticle diffusion rate constant of stage 2; C_2_: intraparticle diffusion constant of stage 2; R^2^_2_: determination coefficient of stage 2; k_p3_: intraparticle diffusion rate constant of stage 3; C_3_: intraparticle diffusion constant of stage 3; R^2^_3_: determination coefficient of stage 3.

**Table 6 ijerph-21-00792-t006:** Comparison of kinetics parameters obtained in studies on BPS adsorption with different adsorbents and in this study.

Model	Adsorbent	Surface Area BET (m^2^/g)	[BPS]_0_ (mg/L)	Carbon Dosage (mg/L)	k_2_ (mg.g^−1^ min^−1^)	q_e_ (exp) (mg.g^−1^)	q_e_ (calc) (mg.g^−1^)	R^2^	Reference
Pseudo-second-order	CCAC	722.22	10–90	200–600	3.74 × 10^−3^	181.98	182.13	0.9954	[23]
	PAC C	726.68	20–60	2	1.3 × 10^−6^	52.2	55.00	0.9900	[40]
	ACSG	1339	20–60	1000	1.12 × 10^−3^	338	334.33	0.9996	[36]
	BETM	880	15	30	5.11 × 10^−4^	351.23	356.80	0.9968	This study
	COCO	673	15	50	1.24 × 10^−3^	133.24	140.70	0.9785	This study
	SIAL	1397	15	30	7.45 × 10^−3^	204.21	205.57	0.9992	This study

**Table 7 ijerph-21-00792-t007:** Parameters obtained after fitting the experimental data to the Langmuir and Freundlich isotherm models.

		BPS			BPA		
Isotherm Models	Parameter	BETM	COCO	SIAL	BETM	COCO	SIAL
	q_max_ (exp) (mg/g)	265.83	129.31	231.12	264.73	122.49	215.94
**Langmuir**	Q^0^_max_ (mg/g)	260.62 ± 4.84	136.51 ± 3.46	248.25 ± 4.49	264.64 ± 4.19	150.03 ± 4.36	231.20 ± 3.44
	k_L_ (L/mg)	5.79 ± 0.89	0.92 ± 0.14	1.06 ± 0.09	2.01 ± 0.24	0.37 ± 0.04	0.87 ± 0.07
	R^2^	0.9887	0.9906	0.9959	0.9940	0.9933	0.9958
	SQR	6.3	1.13	1.24	3.24	0.84	1.58
	R_L_	0.0110	0.0657	0.0574	0.0315	0.1503	0.0656
**Freundlich**	k_F_ (mg/g)/(mg/L)^n^	208.68 ± 2.99	83.17 ± 3.78	147.53 ± 3.22	188.61 ± 4.55	60.38 ± 4.05	132.27 ± 3.20
	n	0.11 ± 8.3 × 10^−3^	0.17 ± 2.2 × 10^−2^	0.20 ± 1.3 × 10^−2^	0.13 ± 1.2 × 10^−2^	0.29 ± 3.2 × 10^−2^	0.20 ± 1.2 × 10^−2^
	R^2^	0.9674	0.8998	0.9774	0.9515	0.9209	0.9706
	SQR	8.4	1.20	1.27	6.82	1.69	1.80

Q^0^_max_: maximum adsorption capacity of the monolayer on the adsorbent; q_max_ (exp): maximum experimental adsorption capacity of the adsorbent; k_L_: Langmuir constant; k_F_: Freundlich constant; n: Freundlich intensity parameter; R^2^: coefficient of determination; k_RP_ and a_RP_: Redlich–Peterson constants; g: exponent of the Redlich–Peterson isotherm; R_L_: separation factor.

**Table 8 ijerph-21-00792-t008:** Comparison of Langmuir isotherm parameters for BPS and BPA adsorptions obtained in different studies, including the present study.

Bisphenol	Adsorbent	k_L_ (L.mg^−1^)	Q^0^_max_ (mg.g^−1^)	R^2^	Reference
BPS	CCAC	0.0086	617.29	0.9983	[23]
BPS	CAP	1.39	83.19	0.9820	[39]
BPA	AC-40	0.64	91.90	0.9960	[15]
BPA	EFB	0.51	41.98	0.9985	[47]
BPS	BETM	5.79	260.62	0.9887	This study
BPS	COCO	0.92	136.51	0.9906	This study
BPS	SIAL	1.06	248.25	0.9959	This study
BPA	BETM	2.01	264.64	0.9940	This study
BPA	COCO	0.37	150.03	0.9933	This study
BPA	SIAL	0.87	231.20	0.9958	This study

## Data Availability

The data are available within the body of the article, and any further data can be requested from authors.

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
