# Peer review of "Removal of Bisphenol S (BPS) by Adsorption on Activated Carbons Commercialized in Brazil"

_ijerph, 2024, doi:10.3390/ijerph21060792_

Round 1

Reviewer 1 Report

Comments and Suggestions for Authors

In this work, the authors focused on the adsorption of organic pollutants using synthesized activated carbons in a batch environment. The study design and properties of the used adsorbent were not performed completely. In addition, experimental data, the presentation, and the discussion are very weak and initial. Therefore, some parts of the paper need to be revised carefully before publication.

1.     The abstract is so general and lengthy, that some quantity results and highlights of the study should be mentioned. It needs to be shortened.

2.     All abbreviations must be checked carefully.

3.     Some typos and grammar errors can be found in the manuscript, and the authors should check the whole manuscript very carefully to avoid any mistakes.

4.     The English language must be improved before publication. The quality of the language is very poor, so it needs to be revised grammatically by a native editor.

5.     Authors have not mentioned the drawbacks of other methods that applied to the removal of organic pollutants, please explain the disadvantages of these techniques (i.e. advanced oxidation processes, bioremediation, ion exchange and etc.). Some references recommended for improving the introduction section Desalination and Water Treatment, 2018, 111, pp. 310–321; Journal of Chemical Technology and Biotechnology, 2016, 91(12), pp. 3000–3010.

6.     COHNS and BET analyses should be performed to determine the textural properties of the used adsorbent and then discussed.

7.     What is the pHZPC of used adsorbent??? The discussion related to the effect of solution pH on the performance of adsorption must be improved using pHZPC of composite.

8.     For all plots, error bars are missed!! Please add error bar values in plots.

9.     Experimental conditions should be given in the caption of the figures.

10.  Thermodynamic studies for the adsorption process must be performed and then discussed. To well support this section, please refer to Korean Journal of Chemical Engineering, 2015, 32(10), pp. 2078–2086.

11.  What is the mechanism of the adsorption process?

12.  The maximum adsorption capacity of Langmuir should be compared with other similar studies.

13.  Reusability tests on the as-prepared adsorbent should be performed.

Comments on the Quality of English Language

The English language must be improved before publication. The quality of the language is very poor, so it needs to be revised grammatically by a native editor.

Reviewer 2 Report

Comments and Suggestions for Authors

There are a few points that the authors may consider.

Introduction – Many papers are available in the scientific literature on the present topics. What is the novelty and originality of this work? Please highlight this.

Line 134 - Have you tried to conduct adsorption tests with real water samples as well (tap, river, lake, and wastewater)? There are so many other organic species that may interact with BPA, completely changing the treatment process.

Line 151 - Why did you use 15 mg/L of BPS? Does it correlate with real conditions?

Line 154 – It is unclear whether you run your experiments under light or dark conditions. Please elaborate.

Line 209 - Surface characterization with SEM-EDS is missing. It is interesting to see the distribution of S in the samples. Please include, if available.

Line 252 - Did you check the effect of pH?

Line 260 – The experiment took 1500 min. Have you run a control experiment with BPA only as well? It is possible to have a significant self-degradation.

Line 419- Can you please elaborate more on the mechanism of the adsorption by the presented materials?

Results – The article contains a lot of data and, therefore, requires better structuring. Please try to merge the figures from section 3, where appropriate.

Line 505 – Present the comparison of your work and BPA/BPS adsorption in the literature in tabular form.

Line 540 – Conclusions are redundant. Please try to focus your attention on the strongest results and explanations.

Line 570 – How practical and scalable is the proposed methodology for real-time applications, and what are the potential challenges in its widespread implementation?

Comments on the Quality of English Language

Some grammatical and syntax errors were observed. Some sentences do not give precise information, and these must be rewritten.

Round 2

Reviewer 2 Report

Comments and Suggestions for Authors

My comments have been fully addressed. Thank you!